# Cost of SARS-CoV-2 self-test distribution programmes by different modalities: a micro-costing study in five countries (Brazil, Georgia, Malaysia, Ethiopia and the Philippines)

Megan A Hansen [1,2] Nkgomeleng A Lekodeba,[3] Joshua M Chevalier [1,2]
Tom Ockhuisen [1] Paula del Rey-Puech,[4] Elena Marban-Castro,[4]
Guillermo Zohar Martínez-Pérez,[4] Sonjelle Shilton [4]
Muhammad Radzi Abu Hassan,[5] Vladimir Getia [6]
Catiuscia Weinert-Mizuschima,[7] Maria Isabelle Tenorio Bezerra,[8] Lensa Chala,[9]
Robert Leong,[10] Remilekun Peregino,[11] Sara Keller,[12] Ineke Spruijt,[12]
Cheryl C Johnson,[13] Sarah J Girdwood,[3,4] Brooke E Nichols[1,2,4]

For numbered affiliations see end of article.

**Correspondence to**
Dr Brooke E Nichols;
brooke.nichols@finddx.org

## ABSTRACT

**Objective** Diagnostic testing is an important tool to combat the COVID-19 pandemic, yet access to and uptake of testing vary widely 3 years into the pandemic. The WHO recommends the use of COVID-19 self-testing as an option to help expand testing access. We aimed to calculate the cost of providing COVID-19 self-testing across countries and distribution modalities.

**Design** We estimated economic costs from the provider perspective to calculate the total cost and the cost per self-test kit distributed for three scenarios that differed by costing period (pilot, annual), the number of tests distributed (actual, planned, scaled assuming an epidemic peak) and self-test kit costs (pilot purchase price, 50% reduction).

**Setting** We used data collected between August and December 2022 in Brazil, Georgia, Malaysia, Ethiopia and the Philippines from pilot implementation studies designed to provide COVID-19 self-tests in a variety of settings— namely, workplace and healthcare facilities.

**Results** Across all five countries, 173 000 kits were distributed during pilot implementation with the cost/ test distributed ranging from $2.44 to $12.78. The cost/ self-test kit distributed was lowest in the scenario that assumed implementation over a longer period (year), with higher test demand (peak) and a test kit price reduction of 50% ($1.04–3.07). Across all countries and scenarios, test procurement occupied the greatest proportion of costs: 58–87% for countries with off-site self-testing (outside the workplace, for example, home) and 15–50% for countries with on-site self-testing (at the workplace). Staffing was the next key cost driver, particularly for distribution modalities that had on-site self-testing (29–35%) versus off-site self-testing (7–27%).

**Conclusions** Our results indicate that it is likely to cost between $2.44 and $12.78 per test to distribute COVID-19 self-tests across common settings in five heterogeneous countries. Cost-effectiveness analyses using these results

## STRENGTHS AND LIMITATIONS OF THIS STUDY

⇒ This research fills gaps in the lack of existing data to help understand the cost of provision and use of COVID-19 self-tests, particularly in lower middle-income countries and middle-income countries.

⇒ The results from this study provide a wide range of costing evidence, in a variety of settings and modalities, to governments making decisions regarding the adoption and scale-up of self-test distribution programmes.

⇒ This study is not a time in motion analysis to assess the time spent by each staff cadre on pilot-related activities; therefore, our estimates on the average time spent by staff on pilot activities per month may be overestimated.

⇒ The pilots in Brazil and Malaysia were ongoing during data collection, requiring the extrapolation of figures and use of budgeted expenditure instead of actual expenditure.

will allow policymakers to make informed decisions on optimally scaling up COVID-19 self-test distribution programmes across diverse settings and evolving needs.

## INTRODUCTION

As of July 2023, the COVID-19 pandemic has led to nearly 768 million cases and 6.9 million deaths worldwide.[1] While testing has been acknowledged as a critical tool to combat the spread of SARS-CoV-2, access to testing remains varied nearly 3 years into the pandemic. In the early stages of the COVID-19 pandemic, polymerase chain reaction (PCR) tests were the preferred method

for the diagnosis of SARS-CoV-2. However, as more cases emerged, and transmission became widespread, low-cost antigen-based rapid diagnostic tests (Ag-RDTs) have increasingly been used to expand access to and uptake of testing.[2]

Despite the benefits, many programmes found limiting the use of Ag-RDTs to professional administration and use still missed many people in need of testing services.[3] The use of self-testing for Ag-RDTs, where individuals perform and interpret their own diagnostic test, is a mechanism to relieve this human resource constraint and rapidly expand potential testing capacity. These more affordable self-tests also allow individuals to test themselves, promptly learn their results and take actions that could improve individual health outcomes and/or prevent onward transmission. For these reasons, COVID-19 self-tests have become increasingly popular in high-income countries (HICs) and were recommended by the WHO in 2022.[1]

Distribution of self-test kits would allow for the scale-up of COVID-19 testing in limited-resource settings, where regular screening through provider-initiated and/or laboratory-based platforms, such as PCR tests, is constrained. While there are some studies on the use of Ag-RDT self-tests in schools, workplaces and within the broader community, these studies focus primarily on the usability or accuracy in comparison with provider-offered testing.[4–7] There is limited research on the relative programmatic costs for the implementation and distribution of COVID-19 self-tests through different modalities, especially in low-resource settings or among communities with restricted access to conventional COVID-19 testing, where self-testing may be of most benefit.[8]

Most of the empirical evidence on the cost of self-testing programmes originates from the human immunodeficiency virus (HIV) self-testing literature,[6 9 10] and more recently from hepatitis C virus (HCV) self-testing.[11] Studies have demonstrated the relative cost of different methods for rollout of HIV self-testing and found, broadly, that there are multiple ways of successfully distributing HIV self-testing, depending on the budget and desired outcome. For HIV self-testing, expansion of testing beyond primary healthcare clinics yielded a net increase in life years saved across all distribution models.[9] Workplace testing was found to be a very cost-effective modality for HIV self-test distribution. It may be plausible that self-testing for COVID-19 is similarly cost-effective in the workplace.

The HIV literature gives us insights into relative differences in cost of implementation of self-test distribution modalities, as well as which modalities are acceptable, appropriate and perceived to be convenient for a variety of populations. However, there are differences in modalities and frequency of testing required for HIV or HCV compared with a respiratory virus such as COVID-19. Given these key differences, more evidence is needed to understand the costs related to delivering COVID-19 self-testing through different modalities for future

planning—for potential future COVID-19 waves, for other seasonal viruses such as influenza and possibly for future outbreaks caused by emerging diseases.

Despite the recently popular and wide use of COVID-19 self-testing in countries with available public health resources, there has been a paucity of data generated to understand the cost of provision, particularly in lower middle-income countries (LMICs) and middle-income countries (MICs). We aimed to determine the cost per COVID-19 self-test allocated by distribution modality as well as the cost per COVID-19 self-test performed. The overarching aim was to provide costing evidence to governments making decisions regarding the adoption and scale-up of self-test distribution programmes to monitor and effectively improve case detection of COVID-19.

## METHODS
### Overview
COVID-19 self-testing pilot studies were implemented in three upper middle-income countries: Brazil, Georgia and Malaysia, as well as one LMIC (the Philippines) and one low-income country (Ethiopia). These pilot implementation studies were designed to provide COVID-19 self-tests to employees, their household members and clinic clients in a variety of different settings—ranging from healthcare settings, to workplaces, to schools (table 1). The goal of the pilot studies was to assess and continuously improve COVID-19 self-testing distribution strategies, the feasibility of the implementation and participants' acceptability—as an appropriate intervention to increase uptake of testing and detection of cases of COVID-19 in the target populations.[7] Additionally, these pilot studies aimed to calculate the cost of providing COVID-19 self-testing via several distribution modalities in different countries.

### Study setting and scope
This is a multisite, prospective cohort study across five countries and multiple workplaces and health facilities (table 1).

### Costing inputs and data collection
We estimated economic costs from the provider perspective using data collected between August and November 2022 in Brazil, Georgia, Malaysia, Ethiopia and the Philippines. All the costs were either collected in US$ or the countries' local currency and converted to 2022 US$ using the average exchange rate for the period January–October 2022 sourced from the country's central/national banks.[12–17]

The costing approach used was a combination of expenditure analysis in estimating financial costs and bottom-up costing to identify any items not included in pilot expenditure records.[18] Cost data were collected across all countries and modalities using a standardised Excel data collection tool that categorised the costs as follows: training, test procurement, distribution and

**Table 1** Description of distribution models

| Country | Distribution modality* | Pilot implementation time period | Potential participants and channel | Secondary distribution to household members | Description |
|---|---|---|---|---|---|
| Brazil (Pelotas, Rio Grande do Sul) | Staff at a workplace of a private medical supply factory for primarily off-site testing | August 2022–February 2023 | Primary distribution to 358 employees | Yes | For the first month, four self-tests were distributed per participant to perform weekly testing. Afterwards, up to six self-test kits were distributed to staff members per month, with recommended use of one per week or at the onset of any COVID-19 symptoms. Results were recorded through a digital platform used at the workplace (mobile application) and participants were expected to seek confirmatory testing, following national guidelines. |
| Brazil (Afogados da Ingazeira, Pernambuco) | Professionals at sites in both the public healthcare sector (20) and the public education sector (14) for primarily off-site testing | October 2022–March 2023 | Primary distribution to 344 employees working in healthcare and 855 employees working in the education sector (up to 1200 total participants) | Yes | During the first month, employees were encouraged to perform one self-test per week regardless if COVID-19 symptoms were present or not (resulting in approximately four tests per month), and an additional self-test is recommended if symptoms arise throughout the week. Afterwards, employees were encouraged to self-test in case of symptoms or contact with a positive case. Results were recorded through a survey link sent via Google Form and participants were expected to seek confirmatory testing, following national guidelines. |
| Georgia (Tbilisi, Kutaisi and Svaneti region) | Staff at schools in a remote mountainous region (24), nursing homes (2) and healthcare centres (hospitals (2) and clinics (1)) for primarily off-site testing | June–December 2022 | Primary distribution to 2066 employees | Yes | Primary self-test kit distribution to staff members at pilot sites for weekly testing, as part of the mandatory national testing programme, and testing in case of symptoms or contact with a positive case, as well as secondary self-test kit distribution to staff members' household members for use in the case of developing symptoms or having a close contact with a positive COVID-19 case. Results were reported to designated staff right after use to be entered in the national database. Positive staff members were linked to care. |
| Malaysia (Alor Setar, Sungai Petani and wider state of Kedah) | Staff of private manufacturing companies (4) for primarily off-site self-testing | November 2022–April 2023 | Primary distribution to up to 2000 employees | Yes | Initial distribution of two self-test kits per staff and household member for self-testing in case of symptomatic/contact/other circumstances, following national guidelines. Kits were refilled on an as-needed basis. Participants received training and information materials on-site prior to test distribution. Participants were asked to report test use and results by Google Form. |

Continued

**Table 1** Continued

| Country | Distribution modality* | Pilot implementation time period | Potential participants and channel | Secondary distribution to household members | Description |
|---|---|---|---|---|---|
| Ethiopia (Addis Ababa and regional states of Amhara and Oromia) | Clients at clinics for on-site self-testing (2) | September–November 2022 | Primary distribution to clients eligible for self-testing when visiting the clinics (total tests available: 1250) | No | Clients arrived at clinic to seek healthcare and were screened for eligibility for partaking in the pilot. If the client was eligible, a self-test kit was provided for on-site self-testing. Clients performed test, interpret and read results while supervised by nurses. If the self-test was positive, the client was sent for confirmatory testing. |
| Philippines (Biñan, Los Baños and wider province of Laguna) | Staff at workplace clinics for on-site testing (7) | September–December 2022 | Primary distribution to workers working at partner clinics (total tests available: 500/site/month) | No | Distribution of self-test kits at partner workplace, with on-site assigned facility to be its clinic. Distribution agents demonstrated self-test kit use to groups. Participants were tested on-site by a provider under supervision of clinic staff. |

*Off-site testing: refers to self-testing outside the workplace (eg, at home); on-site testing: refers to self-testing at the workplace.

storage, staff costs (management and service delivery), data reporting, supplies and consumables, and communication (table 2; details for each country and modality in online supplemental apendices A1–A6). Costs in Brazil, Georgia and Malaysia were collected while our team travelled to the sites and investigated financial records. Costs in Ethiopia and the Philippines were reported by our in-country research partners.

Shared costs were allocated to each distribution modality using the proportion of tests distributed and the proportion of staff for costs driven by this (for example, training costs). Staff salaries were obtained from the in-country implementing partners and allocated by the proportion of time spent on COVID-19 self-test distribution as estimated by in-country implementing partners. Where appropriate, staff salaries for the equivalent staff cadre within the Ministry of Health or government were used in order to better align with routine implementation assumptions in the future.

### Scenario analysis
We calculated the total cost and the cost per self-test kit distributed for three main scenarios, shown in table 3, that differed depending on the costing period used (actual pilot period compared with a theoretical 1-year implementation period), the number of tests distributed (actual, planned, scaled assuming an epidemic peak), the price of the self-test kit (pilot price and a reduction of 50%), and a final scenario considering scaled-up peak implementation with the exclusion of staffing costs assuming that self-test kit distribution would be integrated with other routine services and the staff costs absorbed by the respective provider (workplace, healthcare facility). We included a 50% reduction in the purchase price of

the tests for scenarios 2 and 3 with the expectation that with possible increased testing volumes, the cost of the test would reduce, as seen with other diagnostic tests.

### Patient and public involvement
Participants were involved in the design, conduct and reporting of the research through inclusion in training, enrolment sessions and reporting of data through various platforms as described in table 1.

### RESULTS
Across all five countries, there were a total of 173 000 test kits distributed during the pilot period (table 4). Georgia had the largest pilot programme (90 000 tests distributed) and Ethiopia the smallest (158 at the two healthcare facilities costed (this was scaled back from a planned distribution of 1250 kits)). The cost per test distributed ranged from $2.44 (Malaysia) to $12.78 (Ethiopia). Despite having the lowest price per self-test kit purchased ($1.00), Ethiopia had the highest cost per test distributed ($12.78). This was due in part to low test volumes and higher resource intensity (ie, staff time) associated with on-site testing at a healthcare facility. Malaysia, in comparison, had the lowest cost per test distributed of $2.44, followed by public workplaces in Afogados da Ingazeira, Brazil ($2.82).

The cost per self-test kit distributed across all countries decreases with a routine implementation period of a year (scenarios 2 and 3) and with an increase in the scale of the programme (scenario 3) (table 4). The cost per self-test kit distributed, while still considering staff and reporting costs, is lowest in scenario 3 with higher test demand (during peaks of incidence) assumed, and a test

**Table 2** Cost categories and description

| Cost category | Description and explanation |
|---|---|
| Test procurement | ► Cost of COVID-19 self-test kit unit<br>► Shipping, customs clearance, duties, taxes |
| Test distribution | ► Transportation cost (eg, fuel) for the transportation of COVID-19 self-test kits to sites (including transport to central storage location if relevant) |
| Test storage | ► Purchase or rental of storage space for COVID-19 self-test, such as a warehouse or storage unit<br>► Purchase of any necessary accompanying supplies used for storage (eg, shelves and boxes) |
| Training | ► Training sessions of staff who are members of the team that are dedicated to the COVID-19 self-test activities<br>► Training sessions for the participants on how and when to self-test, how to report the results and where to find more information about self-testing<br>► Design and printing of training materials<br>► Venue, catering, supplies and transport costs associated with training |
| Communication | ► Designing and ordering advertisement posters and stickers<br>► Designing and printing information materials, such as letters and pamphlets<br>► Designing and distributing digital communication materials (eg, instructional videos, digital infographics)<br>► Any additional communication activities that may accrue cost, such as awareness training<br>► Any internet or mobile-associated communication costs |
| Staff: management | ► Cost of staff time spent on activities to manage the COVID-19 self-testing campaign<br>► Cost of time spent on project implementation and support (including any consultant's time where appropriate) |
| Staff: service delivery | ► This includes the salary of the staff, and the average proportion of their time spent dedicated to COVID-19 self-test operations<br>► Activities include staff time spent on:<br>○ Collecting COVID-19 self-tests to bring to appropriate testing facility (eg, workplace, clinic)<br>○ Storing and organising COVID-19 self-tests<br>○ Enrolment sessions for initiation of COVID-19 self-test distribution to participants<br>○ Assisting or administering the COVID-19 self-test to participants<br>○ Training participants on how to use the COVID-19 self-test, including information and sensitisation activities, answering questions during the pilot |
| Data and reporting | ► Development and use of a platform where results are documented and reported<br>► Record keeping after COVID-19 self-tests are distributed, maintaining registers, data compilation and entry<br>► Analysing reported records and completing reports<br>► Technical working group or other meetings to discuss and report on data |
| Supplies and consumables | ► Any infection, prevention and control materials used (masks, hand sanitisers, gloves, alcohol, biohazard bags) for on-site testing or training<br>► Any office supplies used for recording results, etc (pens, paper) |

**Table 3** Three different implementation scenarios: three different implementation scenarios with different subscenarios and other variables impacting the cost analyses

| Scenario | Scenario description | Period | Number of tests | Cost |
|---|---|---|---|---|
| 1 | Pilot costing | Pilot period (preparation time+implementation) | Actual distribution as of 30 November 2022 extrapolated for the remaining period of the pilot | Pilot test kit cost |
| 2 | Routine implementation on annual basis | Annual (12 months) | Planned distribution extrapolated to 12 months | 50% reduction in the price of the test kit |
| 3 | Scaled-up peak implementation* | Annual (12 months) | Doubling of planned annual test distribution to accommodate an epidemic peak period (increased testing for the same number of beneficiaries) | 50% reduction in the price of the test kit, and 50% reduction in the price of the test kit with no data platform costs and no staff costs† |

*Assumed that staff did not have to increase to facilitate a doubling of tests distributed.
†Assumed that staff and data/reporting costs would be absorbed by the respective provider (workplaces or health facilities) and self-test kit delivery integrated into other activities and existing health data platforms.

**Table 4** Programme characteristics and all-inclusive cost per test for each country during the pilot (scenario 1) and annual routine periods of implementation (scenarios 2 and 3)

| Country and distribution method | Scenario 1—pilot implementation | | Scenario 2—routine implementation | | | Scenario 3—scaled peak implementation* | | |
| --- | --- | --- | --- | --- | --- | --- | --- | --- |
| | Total tests distributed with purchase price of self-test kit | Cost per test: all-inclusive | # of tests distributed | Cost per test: using current self-test kit price | Cost per test: using reduced self-test kit price | Cost per test: using current self-test kit price | Cost per test: using reduced self-test kit price | Cost per test: with integrated staff, data and reporting |
| Brazil (Pelotas) ▶ Primarily off-site ▶ Unassisted | **10 000 @ $4.50** | $7.38 | 20 000 | $6.54 | $4.29 | $5.62 | $3.37 | $2.55 |
| Brazil (Afogados) ▶ Primarily off-site ▶ Unassisted | **26 000 @ $2.00** | $2.82 | 52 000 | $2.59 | $1.59 | $2.30 | $1.30 | $1.05 |
| Georgia ▶ Off-site ▶ Unassisted | **90 000 @ $2.00** | $4.44 | 300 000 | $3.34 | $2.34 | $2.95 | $1.95 | $1.61 |
| Malaysia ▶ Off-site ▶ Unassisted | **45 000 @ $1.55** | $2.44 | 135 000 | $2.06 | $1.28 | $1.82 | $1.04 | $0.86 |
| Ethiopia ▶ On-site ▶ Provider assisted | **158 @ $1.00** | $12.78 | 5040 | $4.22 | $3.72 | $3.80 | $3.30 | $1.97 |
| Philippines ▶ On-site ▶ Unassisted | **2000 @ $1.32** | $4.99 | 6000 | $4.51 | $3.85 | $2.92 | $2.26 | $0.95 |

An extended version of this table can be found in the online supplemental appendix A7.
*Twice as many tests are distributed in scenario 3 as in scenario 2 (as also described in table 3).

kit price reduction of 50% ranging from $1.04 in Malaysia to $3.37 in Brazil (Pelotas). The total cost is however high for both scenarios 2 and 3 due to the high procurement costs associated with procuring more tests as well as the longer period of implementation, up to $1.7 million for the largest programme in Georgia. When assuming that self-test kit distribution would become a routinely integrated service with a reduced purchase price, and

assuming zero staffing and data/reporting costs, the cost per test for scenario 3 ranged from $0.86 in Malaysia to $2.55 in Brazil (Pelotas). However, if data and reporting costs were not absorbed by existing health data platforms, data and reporting would only increase the cost per self-test by $0.00–0.12 in Malaysia and Brazil (Pelotas), respectively.

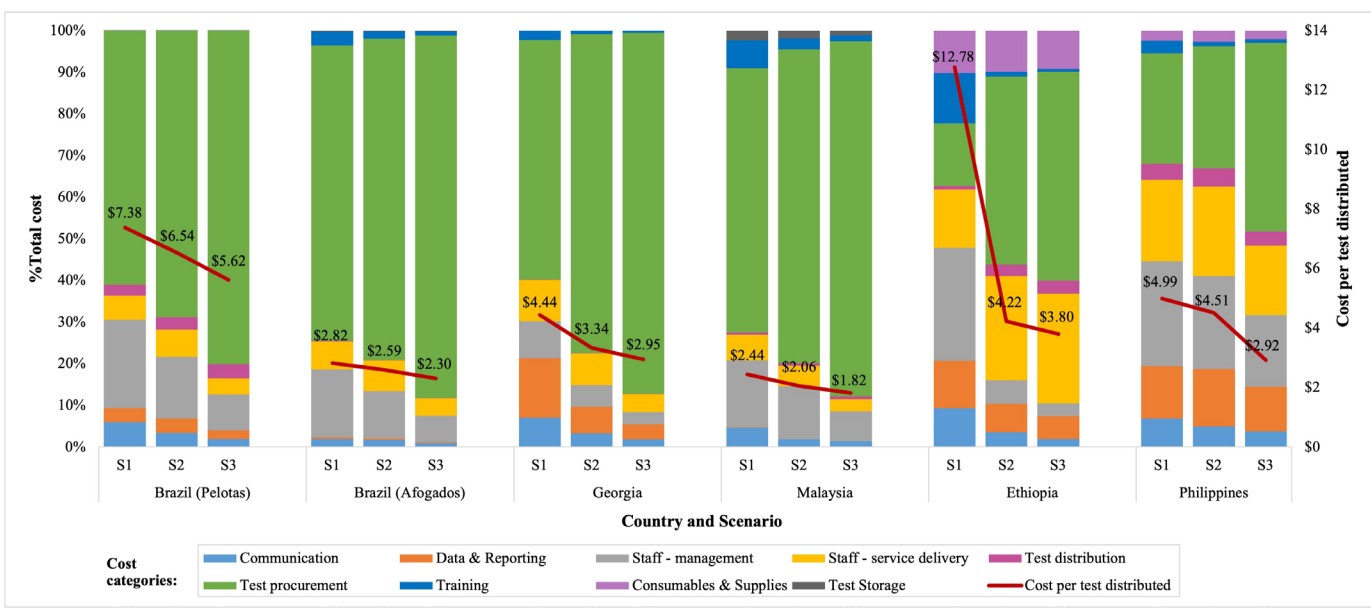

**Figure 1** Cost drivers and the cost per test distributed by country and scenario at current self-test kit procurement prices.

Figure 1 summarises the proportion of total costs that each cost category contributes to across scenarios and countries. Across all countries and scenarios, test procurement (the price of the self-test kits themselves including the costs shipping, etc) contributes the largest proportion and becomes even more important at a greater scale (scenario 3). Test procurement costs range from 58% to 87% for countries with off-site testing (Georgia, Malaysia, Brazil), and 15% to 50% for countries with on-site testing (Ethiopia and the Philippines). Staff cost was the next key cost driver, particularly for distribution modalities that had on-site testing (29–35%) versus distribution modalities with off-site testing (7–27%). Service delivery staffing costs (staff directly involved in the distribution of the self-test kits) ranged from as little as 3% in Malaysia (scenario 3) to 26% in Ethiopia (scenario 3). In the Philippines, Malaysia and both pilots in Brazil, staff management costs were the second largest cost driver ranging from 6% to 25% of total costs but decreasing with the scale of the programme. This is particularly notable in the case of Ethiopia, which saw a sharp drop-off in the contribution of staff management costs when moving from a pilot programme to more routine implementation as fixed management costs are spread across a higher number of tests.

## DISCUSSION

We aimed to determine the cost per COVID-19 self-test allocated by distribution modality as well as the cost per COVID-19 self-test performed. Across all five countries analysed in this study, there were a total of 173 000 kits distributed with the cost per test distributed ranging from $2.44 to $12.78. The cost per self-test kit distributed is lowest in the scenario that assumed implementation over a longer period (1 year), with higher test demand (peak) and a test kit price reduction of 50% ($1.04–3.07).

COVID-19 self-tests are generally cost-saving compared with PCR assays and professional use Ag-RDTs. The average cost of an automated PCR assay is at least $10.00, excluding the cost of sample collection or result delivery.[2] In comparison, the cost of a COVID-19 self-test in the pilot studies ranged from $1.00 to $4.50 making it a more affordable option for expanding access to testing and thereby increasing detection of COVID-19 cases and enacting recommended preventative measures.[2] Despite similarities across countries in terms of income categorisation (three countries are categorised as MICs) as well as distribution modality (five out of the six were workplace distribution models), the cost of *distributing* COVID-19 self-tests varied substantially. This variability in cost can be attributed to the type of COVID-19 self-test that was used (as shown above), the amount of self-tests that were procured, and the modality of distribution or pilot implementation methodology. We found that across countries and scenarios, test procurement (the price of the self-test kit and shipping) constitutes the highest proportion of the total costs of distribution: 58–87% for countries with

off-site testing and 15–50% for countries with on-site testing. While this might suggest that country governments have little flexibility in influencing the overall cost of distributing COVID-19 self-tests outside of procurement negotiations, there are still substantial costs related to the delivery of COVID-19 tests and as such potential for cost reduction, particularly through the design of the distribution modality itself (for example, on-site vs off-site testing). Additionally, the substantial cost of test procurement and shipping may indicate that purchasing COVID-19 self-tests from regional manufacturers or better supply management and coordination may help reduce costs.

We can see that tests that were performed on-site resulted in a greater unit cost per self-test compared with tests that were performed off-site. For example, the on-site pilot implemented in healthcare facilities in Ethiopia ($12.78) required a significantly greater amount of staff time and consumables when compared with the off-site pilot implemented in Malaysia ($2.44). We would expect to see further deviation in the cost of distribution of self-tests in both HICs and LMICs as donor funding and access to Ag-RDTs differ in settings with high and low resources, respectively. Additionally, we can see that the cost to distribute COVID-19 self-tests can be expected to differ during routine programme implementation rather than pilot implementation. We attempted to estimate how start-up costs and pilot staff planning and management costs dedicated to the self-test programme would be reduced during routine implementation (scenarios 2 and 3). This fluctuation in cost can be described as the 'U-shape phenomenon'; the cost per test distributed is initially high but decreases as the programme scales up distribution since fixed start-up (data systems, planning and communication design) and once-off costs (training) are spread over a greater number of tests. Further, we showed how this cost is reduced when the distribution of self-test kits is completely integrated into routine operations and the staffing costs are incrementally small and absorbed by the provider. However, at a certain level of scale, the cost per test will increase again as diseconomies of scale initiate as well as when additional staff and training are required to facilitate the distribution of more and more tests.

Prior to this study, little was known about the cost to implement COVID-19 self-testing programmes; however, self-test distribution has been offered via different modalities for various infectious diseases, including HIV, in which we can draw some comparisons. Sande *et al* found that the key cost drivers of offering HIV self-test kits at public health facilities across four sub-Saharan African countries were personnel and the cost per test kit.[6] This was largely true across distribution modalities and countries for COVID-19 self-testing, as test procurement comprised the largest contribution to total costs, followed by staff management and service delivery categories, even though the cost per test procured differed by country. In the analysis by Sande *et al*, they observed economies of

scale across sites, as the cost per HIV self-test kit distributed was lower among sites that distributed more test kits, with the cost ranging from $4.27 to $13.42.[6] We observed a similar trend in our analysis when scaling up the number of COVID-19 self-test kits distributed, assuming staff costs remain consistent, and start-up related costs begin to taper off, with increased distribution. In another analysis by Matsimela *et al* who looked at HIV self-test kit distribution across 11 modalities in South Africa, including workplaces, they observed economies of scale related to test kit distribution volume. Test kit cost and personnel were indeed the greatest cost drivers in the workplace modalities in that analysis, with the cost to distribute an HIV self-test in the workplace ranging from $5.30 to $6.29.[10] While we cannot directly compare the cost to distribute an HIV self-test kit with a COVID-19 self-test kit due to the inherently different nature of the two viruses' epidemiology (COVID-19 can require more frequent repeat testing), as well as the nature and stigmatisation of diagnosing HIV, we do see a similar range in the cost per COVID-19 self-test kit distributed from $2.44 to $12.78. Our analysis provides an overview of the potential overall costs to offer COVID-19 self-testing programmes.

To our knowledge, this study is the first to estimate the costs of COVID-19 self-test distribution across different countries and settings. We determined the costs based on the best available evidence from pilot studies conducted in five heterogeneous countries across predominately workplace distribution modalities. There are however several limitations. First, while we tried to exclude study-specific costs, there might be protocol-induced higher resource use costs—which may not be observed at scale-up (including higher uptake of testing and reporting due to pilot-specific follow-up). This might result in higher total costs, but also more tests performed potentially resulting in a net effect on the cost per test. Additionally, we did not conduct a time in motion analysis to assess the time spent by each staff cadre on pilot-related activities. As such, it is probable that our estimates on the average time spent by staff on pilot activities per month are overestimated. It is likely that the time spent on test kit distribution will decrease after the starting months, and especially with routine implementation. We tried to adjust for this by excluding all staff costs under scenario 3 to provide a lower bound of what this cost might be, especially with a 50% reduction in the purchase price of the self-tests. Also, in scenario 3, we have assumed that there would be no need for additional staff or staff time if the number of tests distributed was to double. However, since it is possible that we overestimated the average time spent by staff on pilot activities per month, we felt that this assumption was reasonable. In some countries (Brazil and Malaysia), the pilot was ongoing, and we had to extrapolate figures (actual COVID-19 test distribution), as well as use budgeted expenditure instead of actual expenditure. Across countries, actual distribution was lower than planned distribution due to low COVID-19 prevalence. The stage of the epidemic would affect the uptake of self-tests and subsequently the cost per test. To account for this, scenario 3 assessed a doubling of testing demand as a result of an assumed peak in the epidemic to approximate high demand or uptake phases. Lastly, outside of the workplace distribution model, the only other modality piloted was the on-site testing at healthcare facilities in Ethiopia. As such, we were unable to understand the relative difference in implementation costs across modalities limiting the generalisability of these costs to other distribution modalities.

## Conclusion

Our results indicate that it is likely to cost between $2.44 and $12.78 per test to distribute COVID-19 self-tests across common settings in five heterogeneous countries with differing economical characteristics, health systems governance, and epidemiological and demographic profiles. Additional costing and cost-effectiveness analyses based on these results will allow policymakers to make informed decisions on optimal approaches to scaling up COVID-19 self-test distribution programmes across settings and countries as epidemiology and testing needs continue to evolve.

**Author affiliations**
[1]Department of Global Health, Amsterdam Institute for Global Health and Development, Amsterdam UMC, University of Amsterdam, Amsterdam, The Netherlands
[2]Department of Global Health, Boston University School of Public Health, Boston, Massachusetts, USA
[3]Health Economics and Epidemiology Research Office, Johannesburg, South Africa
[4]FIND, Geneva, Switzerland
[5]Clinical Research Center, Hospital Sultanah Bahiyah, Alor Setar, Malaysia
[6]Lugar Research Center, National Center for Disease Control and Public Health, Tbilisi, Georgia
[7]LIFEMED, Pelotas, Brazil
[8]Secretaria Municipal de Saúde, Afogados da Ingazeira, Brazil
[9]KNCV Tuberculosis Foundation, Addis Ababa, Ethiopia
[10]KNCV Tuberculosis Foundation, Manila, Philippines
[11]The Aurum Institute for Health Research, Accra, Ghana
[12]KNCV Tuberculosis Foundation, Den Haag, The Netherlands
[13]WHO, Geneva, Switzerland

**Acknowledgements** We would like to thank all the in-country study teams and implementing partners for their assistance with acquiring data: Brazil (Antakly: Gisela Antakly and Joselito Pedrosa and Germsure: Ana Flavia Pires and Cassia Gonçalves, Pelotas: Lifemed, Afogados da Ingazeira: Municipal Health Secretary); Malaysia (FIND: Xiao Hui Sem and Farhana Zulkifli; MOH of Malaysia: Dr Sunita Binti Abdul Rahman and Dr Chan Huan-Keat); Ethiopia (KNCV: Lensa Chala and Israel Mitiku); Philippines (KNCV: Robert Leong); Georgia (FIND: Maia Japaridze and Ia Jikia; National Center for Disease Control); and all sites participating in the pilots' implementation.

**Contributors** Guarantor – MAH, SJG, BEN. Conceptualisation—BEN. Data curation— MAH, NAL, JMC, TO, PdR-P, EM-C, GZM-P, LC, RL and SJG. Formal analysis—MAH, NAL, JMC, TO and SJG. Investigation—MAH, NAL, JMC, TO and SG. Methodology—MAH, NAL, JMC, TO and SG. Supervision—SG and BEN. Writing (original draft)—MAH, NAL, JMC, TO, SJG and BEN. Writing (review and editing)— MAH, NAL, JMC, TO, PdR-P, EM-C, GZM-P, SS, MRAH, VG, CW-M, MITB, LC, RL, RP, SK, IS, CJ, SJG and BEN.

**Funding** This costing study was funded by KfW Group, the German state-owned investment and development bank (grant number: KfW-TBBU02).

**Disclaimer** The funders played no role in the design of the study; the collection, management, analysis or interpretation of data; writing of the report; or the decision to submit the report for publication.

**Competing interests** None declared.

**Patient and public involvement** Patients and/or the public were involved in the design, or conduct, or reporting, or dissemination plans of this research. Refer to the Methods section for further details.

**Patient consent for publication** Not applicable.

**Ethics approval** This study involves human participants. The Georgia study was granted ethical approval by the National Center for Disease Control (NCDC) and Public Health Institutional Review Board in Tbilisi, Georgia (ref no: 2022-049, 24 May 2022). The Malaysia study was ethically approved by the Medical Research and Ethics Committee (MREC), Ministry of Health Malaysia in Putrajaya, Malaysia (ref no: NMRR ID-22-02021-UCR, 26 October 2022). The pilot in Ethiopia was granted ethical approval by the Ethiopian Public Health Institute Institutional Review Board (EPHI-IRB) in Addis Ababa, Ethiopia (ref no: EPHI-IRB-421 2022, 19 August 2022). The study in the Philippines was approved by Saint Cabrini Medical Center–Asian Eye Institute Ethics Review Committee (SCMC–AIE ERC) in Manila, the Philippines (ref no: QR-ERC-002/05/00/12032020, 19 October 2022). Based on the provisions of Chapter IX, items I and IX, of Resolution 674 of 2022, approval from the Research Ethics Committee/National Research Ethics Council (CEP/CONEP System) was waived for the pilot projects in Pelotas and Afogados da Ingazeira, Brazil, as the programme only used public domain data, did not identify the participants and was conducted alongside routine operations at the targeted workplaces. Furthermore, to ensure the safety of individuals enrolled in the programme was carried out considering all ethical principles defined in Brazilian legislation on the General Personal Data Protection Law (LGPD). All welcomed employees voluntarily signed the free and informed consent form (ICF) after agreeing to participate in the programme. The socioeconomic evaluation was designed, conducted and reported in accordance with consolidated criteria for reporting health economic evaluations.

**Provenance and peer review** Not commissioned; externally peer reviewed.

**Data availability statement** Data are available upon reasonable request. The data used and analysed for this work is readily available in the appendix.

**ORCID iDs**
Megan A Hansen http://orcid.org/0009-0007-0003-7179
Joshua M Chevalier http://orcid.org/0000-0001-8873-6669
Tom Ockhuisen http://orcid.org/0009-0002-3773-3286
Sonjelle Shilton http://orcid.org/0000-0003-1569-8758
Vladimir Getia http://orcid.org/0000-0001-9337-1908

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
