## [Reviewer comments · BMJ Open]

ARTICLE DETAILS

TITLE (PROVISIONAL)	The cost of SARS-CoV-2 self-test distribution programs by different modalities, a microcosting study in five countries: Brazil, Georgia, Malaysia, Ethiopia, and the Philippines
AUTHORS	Hansen, Megan; Lekodeba, Nkgomeleng; Chevalier, Joshua; Ockhuisen, Tom; del Rey-Puech, Paula; Marban-Castro, Elena; Martínez-Pérez, Guillermo; Shilton, Sonjelle; Radzi Abu Hassan, Muhammad; Getia, Vladimir; Weinert-Mizuschima, Catuscia; Tenorio Bezerra, Maria Isabelle; Chala, Lensa; Leong, Robert; Peregino, Remilekun; Keller, Sara; Spruijt, Ineke; Johnson, Cheryl; Girdwood, Sarah; Nichols, Brooke E.

VERSION 1 – REVIEW

REVIEWER	Pandey Santosh Iowa State University, Electrical and Computer Engineering
REVIEW RETURNED	11-Oct-2023

GENERAL COMMENTS	The work presents a study on the cost of COVID-19 self test kits in 5 different locations across four different countries. The compilation of the data and subsequent data analysis is impressive and appealing to the field of diagnostics. Overall the manuscript is very well written and clear. There are some minor comments and suggestions: 1) The full name of Ag-RDTs is not mentioned (Line 71). Please add that.2) Maybe I missed the point where the data in Table 1 and Supplemental File was collected from in the 5 regions. Was it provided by the government or some monitoring body? Maybe a subsection on 'data collection' would be advisable.3) Was any data collected on PCR tests for a comparison with the Ag self test kits? Were there other at-home self test kits used in these geographic regions besides the ones listed in the Supplemental File? Because of having multiple test kit options to end-users, the cost of one particular test kit may vary depending on the market competition.4) In Table 2, the cost categories (such as distribution, training etc.) would vary depending on the economics and financial health of the country under study, especially in low-income countries where labor costs are much cheaper than high-income countries. How did you compare the relative costs in different countries?5) It is suggested to include examples of commercial COVID-19 self test kits to describe the available options for patients during that period. Accuracy, ease of use, cost, and availability are some of the important deciding factors for adoption of a certain COVID-19 test kit. A review paper describes these examples of test kits: Benda et al, Sensors, 2021, https://doi.org/10.3390/s21196581
--

REVIEWER	Rebecca K. Green PATH
REVIEW RETURNED	19-Jan-2024

GENERAL COMMENTS	A well-written paper and concise presentation of a lot of work! This is an important contribution to the literature as countries transition to learning to navigate COVID waves and figuring out what public health should look like to handle COVID. A few questions, comments, and suggestions for consideration: Scenario analysis (p. 10):  - Why was a 50% reduction used? Suggest adding a sentence or two for justification. - I appreciate this analysis to exclude staffing costs associated with distribution to closer align with a real-world scenario, did you consider excluding costs associated with a COVID-specific data platform as well, as that would theoretically integrate into existing health data platforms? - It's unclear to me in Table 3 whether Scenario 3 considers the 50% reduction in test kit price and the removal of staffing costs separately or together.  o Similarly in Table 4, does the Integrated Staff column include the reduced test kit price or is that cost per test based on the current test kit price? o If each factor is considered singularly, it would be interesting to see what considering both at the same time does to the cost, since they shouldn't be mutually exclusive and could co-occur. Aesthetic suggestions for Figure 1:  - Remove the grid lines - Use a solid fill color - Match the font to the figure caption/main text - Increment your y-axis by 20% instead of 10% to make it less squished - Remove the country label from S1 and span it across S1-S3 to fix the text wrapping issue - Remove the price labels on each bar (you have a dual axis, so it's easy enough for the reader to see approximate test cost by the trend line and it's redundant to the information in Table 4) CHEERS Checklist, Line 10: Discount rate is not the same as currency conversion rate, this should be NA. A discount rate is typically used in cost effectiveness analyses to properly value future costs and benefits associated with health interventions given that people value future effects less than present effects. Here's a nice summary of the practice: https://onlinelibrary.wiley.com/doi/full/10.1111/j.1524-4733.2004.74002.x . Minor suggestions to improve readability:  - P. 6, Line 55 (within table): contact (not contract) - Line 269: remove "even" and "further"
--

VERSION 1 – AUTHOR RESPONSE

Comments from Reviewer 1 (Dr. Santosh Pandey, Iowa State University):

Comment 1: The full name of Ag-RDTs is not mentioned (line 71). Please add that.

Response 1: We appreciate the reviewer for catching this error, you can now see that the full name of "Ag-RDTs" has been specified as "antigen rapid diagnostic tests" in line 84 of the manuscript.

Comment 2: Maybe I missed the point where the data in Table 1 and Supplemental File was collected from in the 5 regions. Was it provided by the government or some monitoring body? Maybe a subsection on 'data collection' would be advisable.

Response 2: We thank the reviewer for their comment; however, line 154 in the manuscript reflects the data collection process:

“The costing approach used was a combination of expenditure analysis in estimating financial costs and bottom-up costing to identify any items not included in pilot expenditure records.18”

The data collection process is also further explained in line 159 of the manuscript:

“Costs in Brazil, Georgia, and Malaysia were collected while our team travelled to the sites and investigated financial records. Costs in Ethiopia and the Philippines were reported by our in-country research partners.”

With that said, we can now see that it may be difficult to find the details regarding data collection in the text. To highlight the data collection process, the header for this section has been changed from “Costing input” to “Costing input and data collection” in line 147.

Comment 3: Was any data collected on PCR tests for a comparison with the Ag self-test kits? Were there other at-home self-test kits used in these geographic regions besides the ones listed in the Supplemental File? Because of having multiple test kit options to end-users, the cost of one particular test kit may vary depending on the market competition.

Response 3: No data was collected on PCR tests as they are not a true comparison for self-tests. Additionally, rather than reporting costs for each country with multiple self-test kit options, our results report costs disaggregated by cost type so the reader can adjust the cost of the test itself to make it more relevant to the current pricing in their respective country.

Comment 4: In Table 2, the cost categories (such as distribution, training etc.) would vary depending on the economics and financial health of the country under study, especially in low-income countries where labor costs are much cheaper than high-income countries. How did you compare the relative costs in different countries?

Response 4: We appreciate the reviewer for raising this concern. The costs outlined in each of the categories described in Table 2 are reported in the Appendix for each respective country. Regarding labor costs, line 164 of the manuscript states that “Staff salaries were obtained from the in-country implementing partners and allocated by the proportion of time spent on COVID-19 self-test distribution as estimated by in-country implementing partners.” Since these costs are reported by the in-country implementors themselves, the variability in costs depending on country income level is already reflected in the data collected.

The same is true for our assumptions for gathering labor costs for potential future routine implementation, as described in line 166, “Where appropriate, staff salaries for the equivalent staff cadre within the Ministry of Health or government were used in order to better align with routine implementation assumptions in the future.”

Comment 5: It is suggested to include examples of commercial COVID-19 self-test kits to describe the available options for patients during that period. Accuracy, ease of use, cost, and availability are some of the important deciding factors for adoption of a certain COVID-19 test kit. A review paper describes these examples of test kits: Benda et al, *Sensors*, 2021, <https://doi.org/10.3390/s21196581>.

Response 5: Our criteria for adoption of self-test kits included the suggested deciding factors above: accuracy, ease of use, cost, and availability at the time of implementation. Consultation with local stakeholders also guided which self-tests would be used in each country. Participants were not asked to choose between types of tests with different properties or costs. Additionally, as mentioned before our results report costs disaggregated by cost type so the reader can adjust the type of self-test used and its respective cost to tailor overall pricing.

Comments from Reviewer 2 (Dr. Rebecca K. Green, PATH):

Comment 1: Scenario analysis (page 10): Why was a 50% reduction used? Suggest adding a sentence or two for justification.

Response 1: We suspected that with increased testing volume that the cost of the test would reduce, as seen with other diagnostic tests. We have included our reasoning for using the 50% reduction in Line 179 of the Methods and Line 328 of the discussion while considering the limitations.

Line 179 now states, "We included a 50% reduction in the purchase price of the tests for Scenarios 2 and 3 with the expectation that with possible increased testing volumes, that the cost of the test would reduce, as seen with other diagnostic tests."

Line 328 now states, "It is likely that the time spent on test kit distribution will decrease after the starting months, and especially with routine implementation. We tried to adjust for this by excluding all staff costs under Scenario 3 to provide a lower bound of what this cost might be, especially with a 50% reduction in the purchase price of the self-tests."

Comment 2: I appreciate this analysis to exclude staffing costs associated with distribution to closer align with a real-world scenario, did you consider excluding costs associated with a COVID-specific data platform as well, as that would theoretically integrate into existing health data platforms?

Response 2: This is a great suggestion; we originally did not consider excluding costs associated with a COVID-specific data platform. We have now updated the scenario that excludes staff costs to exclude data and reporting costs now, with the assumption that these costs would be absorbed by the provider and integrated into existing health data platforms. These changes are now reflected in the descriptions of the Cost column in Table 3 and the data figures in the Integrated Staff and Data/Reporting column of Table 4.

Comment 3: It's unclear to me in Table 3 whether Scenario 3 considers the 50% reduction in test kit price and the removal of staffing costs separately or together.

Response 3: We thank the reviewer for addressing potential confusion in Table 3. We have included the suggestion from Comment 2 above and adjusted the Cost column in Table 3 to indicate that Scenario 1 uses pilot test kit costs, and that scenario 3 has two cost considerations: a) 50% reduction in the price of the test kit, and b) 50% reduction in the price of the test kit with no data platform costs and no staff costs. We hope that this clarifies the costs in Scenario 3 and headings in Table 4.

Comment 4: Similarly in Table 4, does the Integrated Staff and Data/Reporting column include the reduced test kit price or is that cost per test based on the current test kit price?

Response 4: We appreciate the reviewer's question. Originally, the Integrated Staff and Data/Reporting column in Table 4 included the non-reduced cost, with the purpose of showing the

individual contribution of reduction in staff. After careful consideration and deciding to exclude data and reporting costs as suggested in Comment 2, we have decided to use the reduced cost. These changes are now reflected in the data figures in the Integrated Staff and Data/Reporting column of Table 4.

Comment 5: If each factor is considered singularly, it would be interesting to see what considering both at the same time does to the cost, since they shouldn't be mutually exclusive and could co-occur.

Response 5: Thank you for this comment. As mentioned in the responses to Comments 2 and 4, we have decided to use a 50% reduction in cost and no data/reporting costs in the Integrated Staff column. We decided to exclude both, rather than each individually because we assume that in a real-world scenario, both the staff and data and reporting costs would be absorbed into existing activities and platforms. Additionally, we can see from Figure 1 that Data & Reporting (colored in orange) costs contribute a small percentage of the overall costs that go into the unit cost of the self-test in Scenarios 1-3. If a country were to prioritize data and reporting and not integrate their methods into existing systems, it would only increase the cost per self-test by \$0.00 – \$0.12. The data and reporting methods used in Malaysia resulted in no difference in the per self-test cost, while the methods used in Pelotas, Brazil resulted in a \$0.12 difference in the cost per self-test.

Line 229 in the manuscript now identifies this distinction, “When assuming that self-test kit distribution would become a routinely integrated service with a reduced purchase price, and assuming zero staffing and data/reporting costs, the cost per test for Scenario 3 ranged from \$0.86 in Malaysia to \$2.55 in Brazil (Pelotas). However, if data and reporting costs were not absorbed by existing health data platforms, data and reporting would only increase the cost per self-test by \$0.00–\$0.12 in Malaysia and Brazil (Pelotas), respectively.”

Comment 6: Aesthetic suggestions for Figure 1:

- Remove the grid lines
- Use a solid fill color
- Match the font to the figure caption/main text
- Increment your y-axis by 20% instead of 10% to make it less squished
- Remove the country label from S1 and span it across S1-S3 to fix the text wrapping issue
- Remove the price labels on each bar (you have a dual axis, so it's easy enough for the reader to see approximate test cost by the trend line and it's redundant to the information in Table 4)

Response 6: We thank the reviewer for their suggestions on how to improve Figure 1. We have incorporated some of the suggestions, and if not, then we have explained why we kept our stylistic choice below.

- The grid lines have been removed; we agree that this makes the figure feel less busy.
- A solid fill color has been used, we also altered some of the fill colors, so they are more distinct to avoid confusion between cost categories.
- The font in the figure has been updated to match the figure caption and main text to maintain consistency.
- We have chosen to keep the y-axis increase by increments of 10% rather than the suggested 20% to avoid confusion, as some of the cost categories are very small. We believe that 10% increments are a more appropriate fit for the data presented in the figure.
- The country label has been centered below S1–S3 along the x-axis to avoid crowding of text.
- We have chosen to include the price labels on each bar so that the figure provides adequate information to aid in ease potential decision-making processes that the results from this paper, and

specifically this figure, may provide information for. However, we have decided to move the price labels so that they are centered on each bar to reduce clutter within the figure.

Comment 7: CHEERS Checklist, Line 10: Discount rate is not the same as currency conversion rate, this should be NA. A discount rate is typically used in cost effectiveness analyses to properly value future costs and benefits associated with health interventions given that people value future effects less than present effects. Here's a nice summary of the practice:

<https://onlinelibrary.wiley.com/doi/full/10.1111/j.1524-4733.2004.74002.x>.

Response 7: We appreciate the reviewer for catching this error and for the useful resource. The discount rate in line 10 of the CHEERS Checklist has been updated to "Not applicable".

Comment 8: Minor suggestions to improve readability:

- Page 6, line 55 (within table): contact (not contract)
- Line 269: remove "even" and "further"

Response 8: These minor suggestions are appreciated. The typo in Table 1 (on page 7 now) has been corrected to "contact" rather than "contract". We also removed the words "even" and "further" in what is now line 292 of the manuscript to avoid repetitiveness and improve readability.

The sentence, beginning at line 292, now states, "Further, we showed how this cost is reduced when the distribution of self-test kits is completely integrated into routine operations and the staffing costs are incrementally small and absorbed by the provider."

Additional clarifications:

In addition to the comments outlined above, we have corrected all other grammatical and spelling errors to improve clarity and conciseness of the manuscript.

We look forward to hearing back from you regarding the submission to BMJ Open and are available for any other comments or concerns. Thank you.

VERSION 2 – REVIEW

REVIEWER	Rebecca K. Green PATH
REVIEW RETURNED	20-Feb-2024
GENERAL COMMENTS	This looks great! The incorporated changes make this a stronger, clearer paper that will be a good contribution to the literature.